# MaNGO – Adaptable Graph Network Simulators via Meta-Learning

**Philipp Dahlinger**[*] **Tai Hoang  Denis Blessing  Niklas Freymuth  Gerhard Neumann**
Autonomous Learning Robots
Karlsruhe Institute of Technology
Karlsruhe

## Abstract

Accurately simulating physics is crucial across scientific domains, with applications spanning from robotics to materials science. While traditional mesh-based simulations are precise, they are often computationally expensive and require knowledge of physical parameters, such as material properties. In contrast, data-driven approaches like Graph Network Simulators (GNSs) offer faster inference but suffer from two key limitations: Firstly, they must be retrained from scratch for even minor variations in physical parameters, and secondly they require labor-intensive data collection for each new parameter setting. This is inefficient, as simulations with varying parameters often share a common underlying latent structure. In this work, we address these challenges by learning this shared structure through meta-learning, enabling fast adaptation to new physical parameters without retraining. To this end, we propose a novel architecture that generates a latent representation by encoding graph trajectories using conditional neural processes (CNPs). To mitigate error accumulation over time, we combine CNPs with a novel neural operator architecture. We validate our approach, Meta Neural Graph Operator (MaNGO), on several dynamics prediction tasks with varying material properties, demonstrating superior performance over existing GNS methods. Notably, MaNGO achieves accuracy on unseen material properties close to that of an oracle model.

## 1  Introduction

The simulation of complex physical systems is of paramount importance in a wide variety of engineering disciplines, including structural mechanics [1–3], fluid dynamics [4–6], and electromagnetism [7–9]. In particular, simulating object deformations under external forces is essential for applications such as robotics [10–12]. Traditional mesh-based simulations are appealing for such problems due to the accuracy of the underlying finite element method [13, 14]. However, these methods are typically slow and require precise knowledge of the simulation parameters, including material properties of objects.

In contrast, data-driven approaches for simulating complex systems have emerged as a promising alternative to traditional mesh-based simulators [15–17]. Among them, Graph Network Simulators (GNSs) have recently become increasingly popular [18–21, 12, 22]. GNSs encode the simulated system as a graph of interacting entities whose dynamics are predicted using Graph Neural Networks (GNNs) [23]. These models are often orders of magnitude faster than classical simulators [19] while being fully differentiable, making them highly effective for downstream tasks such as inverse design problems [20, 24]. Moreover, these models do not require knowledge of simulation parameters as they directly learn from the training data. However, they must be retrained from scratch for even

---

[*]correspondence to `philipp.dahlinger@kit.edu`

39th Conference on Neural Information Processing Systems (NeurIPS 2025).

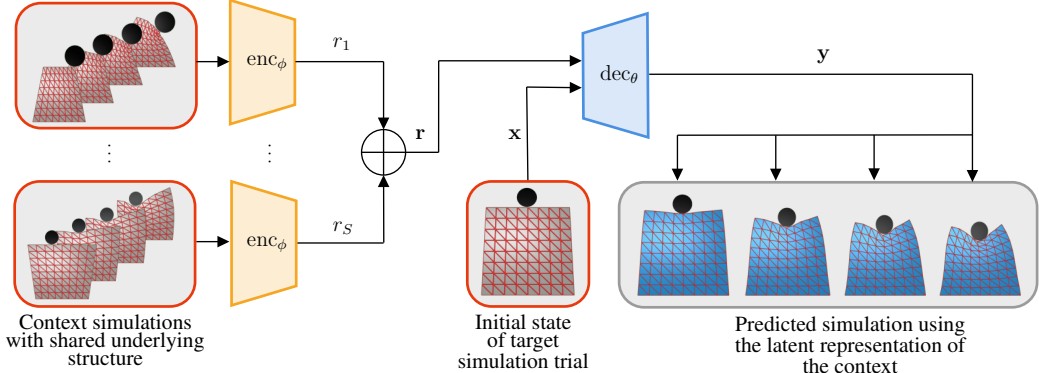

Figure 1: Our proposed Meta Neural Graph Operator (MaNGO) approach: The context set is aggregated to form a latent representation of material properties. Given an unseen initial state, the Graph Network Simulator (GNS) uses this latent representation to generate trials that follow the material laws of the context set, enabling accurate predictions for new conditions. The example prediction aligns perfectly with the ground truth data.

minor variations in physical parameters, and require data collection for each new parameter setting that is often either costly and time-consuming or even impossible.

To overcome this, Sanchez-Gonzalez et al. [25] proposed to use the simulation parameter as conditional information when training the GNS, allowing for generalization to unseen simulation parameters. However, such an approach only works under two assumptions. Firstly, training data must be labeled with the corresponding simulation parameter, and secondly, the simulation parameter must be available at test time for the desired simulation. While the first assumption is mild and often satisfied in simulation settings, the second assumption requires solving the inverse problem of inferring the underlying physical parameters from observed system behavior [26]. Material estimation or system identification can be viewed as a specific instance of this problem [27–32]. However, solving such inverse problems is challenging, as they are typically ill-posed and require explicit knowledge of the governing partial differential equation (PDE) [30, 27, 28].

To overcome this challenge, we investigate data-driven adaptation of GNS – enabling fast and accurate simulations for unknown parameters using only a few simulation trials. Our work builds on the premise that training data from different simulations shares a common 'latent' underlying structure. We aim to learn this structure via meta-learning using Conditional Neural Processes (CNPs). To that end, we propose a novel framework called Meta Neural Graph Operator (MaNGO) that builds on Message Passing Networks (MPNs) and neural operator methods to ensure efficient processing of spatiotemporal data. We validate our approach on several dynamics prediction tasks with varying material properties, demonstrating superior performance over existing GNS methods. Notably, our method achieves accuracy on unseen material properties close to that of an oracle model which has access to the simulation parameters at test time. To summarize, we identify our contribution as follows: (i) we successfully use meta-learning with Conditional Neural Processes (CNPs) for graph network simulators allowing for fast and accurate adaptation to unseen physical parameters. (ii) we identify shortcomings of existing architectures for handling spatiotemporal data and propose a novel GNS architecture. (iii) we provide a set of new benchmark tasks suited for testing the adaptation capability of GNS[2].

## 2 Preliminaries

**Graph and Message Passing Neural Networks.** Graph Neural Networks (GNNs) are a class of neural networks designed to process graph-structured data by iteratively updating node representations through localized message passing. Here, a graph is a defined as $\mathcal{G} = (\mathcal{V}, \mathcal{E}, \{\mathbf{m}_v^0\}_{v \in \mathcal{V}}, \{\mathbf{m}_e^0\}_{e \in \mathcal{E}})$ with nodes $\mathcal{V}$, edges $\mathcal{E}$, and associated vector-valued node and edge features $\mathbf{m}_v^0$ and $\mathbf{m}_e^0$. A Message Passing Network (MPN) [25, 19], a GNN architecture well-suited for graph-based simulations, consists of $K$ message passing steps, which iteratively update the node and edge features based on

---

[2]Code: `https://github.com/ALRhub/mango`   Dataset: `https://zenodo.org/records/17287535`

the graph topology. Each such step is given as

$$\mathbf{m}_e^{k+1} = f_{\mathcal{E}}^k(\mathbf{m}_e^k, \mathbf{m}_v^k, \mathbf{m}_w^k), \text{ with } e = (v, w), \qquad \mathbf{m}_v^{k+1} = f_{\mathcal{V}}^k\Big(\mathbf{m}_v^k, \bigoplus_{e \in \mathcal{E}_v} \mathbf{m}_e^{k+1}\Big), \qquad (1)$$

where $\mathcal{E}_v \subset \mathcal{E}$ are the edges connected to $v$. Further, $\bigoplus$ denotes a permutation-invariant aggregation operation such as the sum, the max, or the mean. The functions $f_{\mathcal{V}}^m$ and $f_{\mathcal{E}}^m$ are learned Multilayer Perceptrons (MLPs), usually with a residual connection. The network's final output are the node-wise learned representations $\mathbf{m}_v^K$ that encode local information of the initial node and edge features.

**Neural Dynamics Prediction.** Our goal is to learn the dynamics of a multibody system, i.e., a trajectory of graphs $(\mathcal{G}^{(t)})_{t \in [0,T]}$ from the initial condition $\mathcal{G}^{(0)}$. We follow Brandstetter et al. [33] and group existing approaches into two categories, *neural operators* and *autoregressive methods*. Neural operator methods treat the mapping from initial conditions to solutions at time $t$ as an input–output mapping learnable via supervised learning. Formally, a neural operator $F_{\mathrm{NO}}$ predicts the graph at any $t \in [0, T]$ from the initial condition, that is, $\mathcal{G}^{(t)} = F_{\mathrm{NO}}\left(t, \mathcal{G}^{(0)}\right)$. In contrast, autoregressive methods, learn to incrementally update the graph starting from $\mathcal{G}^{(0)}$:

$$\mathcal{G}^{(t+\Delta t)} = F_{\mathrm{AR}}\left(\Delta t, \mathcal{G}^{(t)}\right).$$

Here, $F_{\mathrm{AR}}$ is the temporal update and $\Delta t \in \mathbb{R}_{>0}$.

**Meta-Learning and Conditional Neural Processes.** To formalize the meta-learning problem using conditional neural processes (CNPs) [34], we consider a meta-dataset $\mathcal{D} = \mathcal{D}_{1:L}$ consisting of $L$, typically small, task datasets $\mathcal{D}_l = \{\mathbf{x}_i^l, \mathbf{y}_i^l\}_{i=1}^{S_l}$ of size $S_l$. Each task consists of inputs $\mathbf{x}_i^l \in \mathbb{R}^{d_x}$ and corresponding evaluations $\mathbf{y}_i^l \in \mathbb{R}^{d_y}$ of unknown functions $f_l$, that is, $\mathbf{y}_i^l = f_l(\mathbf{x}_i^l) + \epsilon_i$, where $\epsilon_s$ denotes (possibly heteroskedastic) noise. Meta-learning hinges on the idea that tasks share statistical structure, allowing for fast adaptation to a target function $f_*$ based on a small target dataset $\mathcal{D}_* = \{\mathbf{x}_i^*, \mathbf{y}_i^*\}_{i=1}^{S_*}$ of size $S_*$. To leverage this shared statistical structure, Conditional Neural Processes (CNPs) use the meta-dataset $\mathcal{D}$ to learn how to generate a latent representation $\mathbf{r} \in \mathbb{R}^{d_r}$ from a given set of $(\mathbf{x}, \mathbf{y})$-pairs. At test time, this latent representation is generated from the target dataset $\mathcal{D}_*$ enabling generalization to unlabeled inputs $\mathbf{x}_*$ from the target task without requiring weight adaptation. Formally, to generate the latent representation $\mathbf{r}$ for a set $\mathcal{S} = \{\mathbf{x}_i, \mathbf{y}_i\}_{i=1}^S$ with arbitrary size $S$, CNPs use a parameterized encoder with a permutation-invariant aggregation method,

$$\mathbf{r} = \bigoplus_{i \in \{1, \dots, S\}} r_i \quad \text{with} \quad r_i = \mathrm{enc}_\phi(\mathbf{x}_i, \mathbf{y}_i), \qquad (2)$$

with parameters $\phi$ and permutation-invariant aggregation method $\bigoplus$. The latent representation $\mathbf{r}$ is then used to predict the mean and variance of a Gaussian distribution over $\mathbf{y}$, given a new input $\mathbf{x}$,

$$p_\theta(\mathbf{y}|\mathbf{x}, \mathcal{S}) = \mathcal{N}\left(\mathbf{y} | \mathrm{dec}_\theta^\mu(\mathbf{x}, \mathbf{r}), \ \mathrm{dec}_\theta^\Sigma(\mathbf{x}, \mathbf{r})\right), \qquad (3)$$

using a parameterized decoder with parameters $\theta$. For training, the task datasets are further split into context sets $\mathcal{D}_l^c \subseteq \mathcal{D}_l$ to train the CNP on different dataset sizes $S_l^c$ which are used to minimize the negative task log-likelihood

$$\mathcal{L}_l(\phi, \theta) = -\mathbb{E}_{\mathcal{D}_l^c \subseteq \mathcal{D}_l}\left[\sum_{i=1}^{S_l} \log p_\theta(\mathbf{y}_i^l | \mathbf{x}_i^l, \mathcal{D}_l^c)\right] \qquad (4)$$

which implicitly depends on $\phi$ through the encoding of $\mathcal{D}_l^c$. Here, the expectation indicates randomly sampled subsets. Thus, meta-learning aims to maximize the overall task likelihood while conditioning the model on smaller subsets of the data. The complete loss is obtained as $\mathcal{L}(\phi, \theta) = \sum_l \mathcal{L}_l(\phi, \theta)$ and is optimized end-to-end using stochastic gradients with respect to $\phi$ and $\theta$.

## 3   Adaptable Graph Network Simulators via Meta-Learning

Having established the foundations, we now explain how meta-learning with Conditional Neural Processes (CNPs) can be used to make graph network simulators adaptable to novel unseen physical parameters. To that end, we start by formalizing our setup and introducing the meta-dataset in

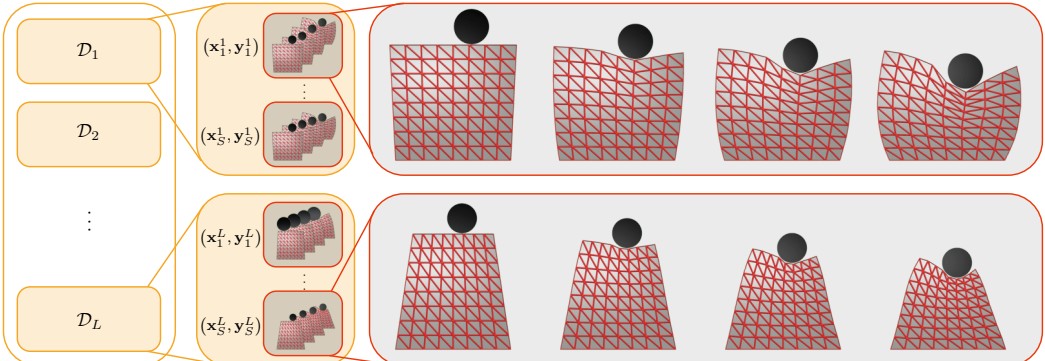

Figure 2: Meta-dataset representation: This figure illustrates the structure of the meta-dataset, consisting of multiple task datasets. Each dataset includes simulations with shared material properties but varying starting conditions. Two example simulations, though starting similarly, produce vastly different results due to their distinct material properties.

Section 3.1. Next, we propose an extension to meta-learning that improves training with additional information that is not available at test-time in Section 3.2. Lastly, we introduce a CNP architecture that is tailored to our setup. Specifically, we propose an encoder that generates a latent representation from spatiotemporal data in Section 3.3. Furthermore, we introduce a novel decoder that addresses shortcomings of existing methods as outlined in Section 3.4.

## 3.1 Problem Formalization and Setup

The aim is to learn a graph network simulator (GNS) that predicts a sequence of graphs $\mathcal{G}_{1:T} \coloneqq \{\mathcal{G}_t\}_{t=1}^T$ describing, e.g., mesh deformation over time without knowing some simulation or physical parameters, which we refer to as $\rho \in \mathbb{R}^{d_\rho}$. For example, $\rho$ could be stiffness or compression properties of a deformable object in the simulation. To that end, we use meta-learning to extract latent representations from similar data to generalize to unseen physical parameters. As outlined in Section 2, in order to perform meta-learning we require a meta dataset $\mathcal{D}$ containing $L$ task datasets $\mathcal{D}_l \in \mathcal{D}$ that in turn consist of multiple input-output pairs $(\mathbf{x}_i^l, \mathbf{y}_i^l) \in \mathcal{D}_l$. In our setting, each output is given by

$$\mathbf{y}_i^l = (\mathbf{p}_{1:T}^l, \mathbf{v}_{1:T}^l), \tag{5}$$

where $\mathbf{p}_{1:T}^l, \mathbf{v}_{1:T}^l \in \mathbb{R}^{T \times N \times d}$ denote the positions and velocities of a sequence of $N$ nodes belonging to (possibly deformable) objects in $d$-dimensional space. An input $\mathbf{x}_i^l$ in our setting is given by

$$\mathbf{x}_i^l = (\mathbf{p}_0^l, \mathbf{v}_0^l, \mathbf{p}_{0:T}^{l,\text{ext}}, \mathbf{v}_{0:T}^{l,\text{ext}}, \mathbf{h}^l), \tag{6}$$

where $\mathbf{p}_0^l, \mathbf{v}_0^l \in \mathbb{R}^{N \times d}$ are the initial node positions and velocities. Here $\mathbf{p}_{0:T}^{l,\text{ext}}, \mathbf{v}_{0:T}^{l,\text{ext}} \in \mathbb{R}^{(T+1) \times N^{\text{ext}} \times d}$ are positions and velocities of 'external' objects, that is, objects that we do not wish to simulate, such as a rigid collider that interacts with the object of interest. Lastly, $\mathbf{h}^l \in \mathbb{R}^{(N+N^{\text{ext}}) \times d_{\mathbf{h}}}$ represents node features that remain constant over time, such as the node's type (deformable or collider) or whether a force is applied to the node. Initial condition $\mathbf{x}^l$ and simulation result $\mathbf{y}^l$ together result in a sequence of graphs $\mathcal{G}_{1:T}^l$ with $N + N_{\text{ext}}$ nodes. In our work, we assume a fixed graph structure: the connectivity and the number of nodes in $\mathcal{G}_t^l$ is constant over time and task datasets $\mathcal{D}_l$. An illustration of a meta dataset $\mathcal{D}$ is shown in Figure 2.

## 3.2 Incorporating Simulation Parameters into Meta-Training

For training, we could follow the procedure outlined in Section 2 and optimize the loss defined in Equation (4). However, contrary to the setup discussed in Section 2, we additionally assume access to the simulation parameters for each training task, i.e., $\{\rho^l\}_{l=1}^L$ but not for target tasks $\mathcal{D}^*$ rendering them useless in the standard meta-learning formulation. Here, we slightly extend the meta training such that we obtain additional learning signals from $\rho^l$. To that end, we use an additional parameterized neural network $f^\psi : \mathbb{R}^{d_r} \to \mathbb{R}^{d_\rho}$ with parameters $\psi$ that aims to predict $\rho^l$ from a

latent representation $\mathbf{r}^l$. Using the following definition for the joint likelihood between $\mathbf{y}$ and $\rho$, i.e.,

$$p_{\theta,\psi}(\mathbf{y}, \rho | \mathbf{x}, \mathcal{S}) = \mathcal{N}\left(\begin{bmatrix}\mathbf{y}\\\rho\end{bmatrix} \middle| \begin{bmatrix}\mathrm{dec}_\theta^\mu(\mathbf{x}, \mathbf{r})\\f^\psi(\mathbf{r})\end{bmatrix}, \begin{bmatrix}\mathrm{dec}_\theta^\Sigma(\mathbf{x}, \mathbf{r}) & 0\\0 & 1\end{bmatrix}\right),$$

we obtain a novel per-task loss function as

$$\mathcal{L}_l(\phi, \theta, \psi) = -\mathbb{E}_{\mathcal{D}_l^c \subseteq \mathcal{D}_l}\left[\sum_{s=1}^{S_l} \log p_{\theta,\psi}(\mathbf{y}^l, \rho^l | \mathbf{x}, \mathcal{D}_l^c)\right],$$

where the full loss again sums over the task losses, $\mathcal{L}(\phi, \theta, \psi) = \sum_l \mathcal{L}_l(\phi, \theta, \psi)$. Intuitively, gradients with respect to the encoder parameters $\phi$ are informed by $\rho$ via $f^\psi$. As another positive side-effect, we obtain an estimate $\hat{\rho} = f^\psi(\mathbf{r})$ which could be used for downstream tasks.

### 3.3 Spatiotemporal Encoder

In Section 2, we treated the encoder and decoder of the CNP architecture as black boxes that are responsible for generating a latent representation and a predictive distribution over outputs, respectively. However, since our data consists of non-standard structures, specifically graphs with both spatial and temporal components, we introduce the following novel architectures.

Recall that the encoder of a CNP generates a latent representation $\mathbf{r}^l \in \mathbb{R}^{d_r}$ from a set of input-output pairs, typically a context set $\mathcal{D}_l^c \subseteq \mathcal{D}_l$. Then, for each $(\mathbf{x}_i^l, \mathbf{y}_i^l) \in \mathcal{D}_l^c$ we separately generate a latent representation, i.e., $r_i^l = \mathrm{enc}_\phi(\mathbf{x}_i^l, \mathbf{y}_i^l)$ which are then aggregated in a permutation-invariant fashion to obtain $\mathbf{r}^l$. As discussed in Section 3.1, each input-output pair in our setting has a spatial and temporal component. We first aggregate over the temporal dimension and then over the spatial dimension. To that end, we combine the inputs $\mathbf{x}_i^l$ (see Equation (6)) and $\mathbf{y}_i^l$ (see Equation (5)) as

$$\mathbf{z}_i^l = (\mathbf{p}_{0:T}^l, \mathbf{v}_{0:T}^l, \mathbf{p}_{0:T}^{l,\mathrm{ext}}, \mathbf{v}_{0:T}^{l,\mathrm{ext}}, \mathbf{h}_{0:T}^l),$$

where $\mathbf{h}^l$ is simply copied for each time step. We ensure translation invariance by subtracting the initial mean position. To remove the temporal component, we apply a 1D convolutional neural network, i.e., $\hat{\mathbf{z}}_i^l = \mathrm{CNN}_{1D}^\phi(\mathbf{z}_i^l)$. Lastly, to obtain a spatial-independent latent representation, we apply a deep set encoder [35] on the node level, that is,

$$r_i^l = \mathrm{enc}_\phi(\mathbf{x}_i^l, \mathbf{y}_i^l) = f_{\mathrm{outer}}^\phi\left(\frac{1}{N + N_{\mathrm{ext}}}\sum_{n=1}^{N+N_{\mathrm{ext}}} f_{\mathrm{inner}}^\phi(\hat{\mathbf{z}}_{i,n}^l)\right),$$

where, $f_{\mathrm{outer}}^\phi$ and $f_{\mathrm{inner}}^\phi$ are neural networks. In this work, we choose deep sets as our spatial aggregation method, as they have demonstrated good performance in graph-level problems similar to ours [36]. Finally, to aggregate a whole set of input-output pairs, we follow Equation (2) and use a permutation invariant aggregation method.

### 3.4 Meta Neural Graph Operator Decoder

We propose a novel MaNGO decoder architecture that combines elements of MeshGraphNet (MGN) [19] and the Equivariant Graph Neural Operator (EGNO) [22]. EGNO predicts graph sequences using equivariant GNN layers and temporal convolutions in Fourier space. While beneficial in some settings, these equivariance constraints can limit performance when not required. We discuss this limitation in Appendix B, where we mathematically show that EGNO struggles on certain tasks. Alternatively, MGN models graph sequences autoregressively using Message Passing Networks (MPNs), which are highly effective for graph-based simulations [19]. However, autoregressive prediction is not ideal when paired with the current meta-learning training setup, see Appendix E for further discussions. To address these limitations, MaNGO retains the strengths of neural operator methods by replacing the equivariant GNN layers with MPNs and equivariant convolutions with a 1D CNN (see Figure 3). Formally, we use the MaNGO decoder to realize $\mathbf{y} = \mathrm{dec}_\theta(\mathbf{x}, \mathbf{r})$ (omitting the task index $l$ for readability). Thus, we define a single MaNGO block as a mapping

$$(\mathbf{m}_{e,1:T}^k, \mathbf{m}_{v,1:T}^k) \mapsto (\mathbf{m}_{e,1:T}^{k+1}, \mathbf{m}_{v,1:T}^{k+1}),$$

following the notation in Section 2. The initial edge and note features $(\mathbf{m}_{e,1:T}^0, \mathbf{m}_{v,1:T}^0)$ are extracted from $(\mathbf{x}, \mathbf{r})$ by copying $\mathbf{r}$ to every node feature and replicating the initial graph $T$ times. Additionally,

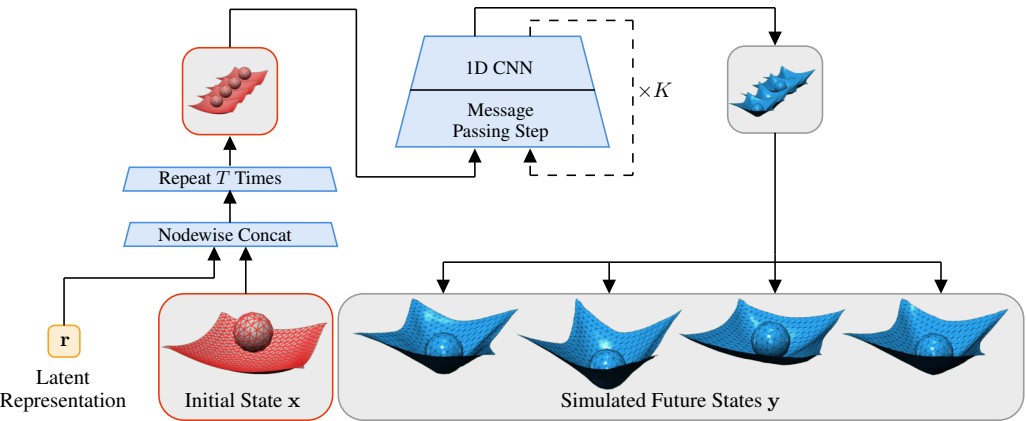

Figure 3: MaNGO Decoder: Our simulator takes a latent representation and an initial state as input. The initial state is combined and iteratively processed to generate a trajectory of graphs. By alternating between a message-passing network for spatial processing and a 1D CNN for temporal processing, the simulator produces accurate dynamic simulations.

we include time embeddings for every time step $t$ and use relative positions as edge features. Further details on the feature creation are provided in Appendix C. As a first step of the MaNGO block, we leverage the MPN for a spatial update, that is, we process the edge and node features according to Equation (1) for each timestep separately to obtain a new tuple $(\mathbf{m}_{e,1:T}^{k+1}, \tilde{\mathbf{m}}_{v,1:T}^{k+1})$. The temporal update is subsequently performed using a 1D residual convolutional layer, that is,

$$\mathbf{m}_{v,1:T}^{k+1} = \mathrm{Conv}_{\mathrm{1D}}^{\theta}(\tilde{\mathbf{m}}_{v,1:T}^{k+1}) + \tilde{\mathbf{m}}_{v,1:T}^{k+1}.$$

After $K$ MaNGO blocks the final node features $\mathbf{m}_{v,1:T}^{K}$ are used for predicting the node positions for every time-step. Specifically, a displacement vector is computed using a parameterized network $f^{\theta}$, that is, $\mathbf{d}_{v,t} = f^{\theta}(\mathbf{m}_{v,t}^{K})$ to obtain node positions $\mathbf{p}_{v,t}$ as $\mathbf{p}_{v,t} = \mathbf{p}_{v,0} + \mathbf{d}_{v,t}$.

The resulting positions $\mathbf{p}_{v,t}$ define the graph sequence $\mathbf{y}$ over time. In this work, we do not predict input-dependent variances, and instead use a fixed $\mathrm{dec}_{\theta}^{\Sigma}(\mathbf{x}, \mathbf{r}) := 1$ to stabilize and simplify the training scheme. If variances are required, for example, for uncertainty estimation, they can be easily predicted from the decoder as well. By alternating between spatial message passing and temporal convolution, the MaNGO simulator efficiently models time-series graph data while avoiding the pitfalls of equivariance over-constraints and autoregressive prediction.

## 4 Related Work

**Learning-based forward simulators.** Using deep neural networks to learn physical simulations has become an emerging research direction in scientific machine learning [19, 37, 38]. Deep learning-based approaches have demonstrated success in applications such as fluid dynamics [39, 40], aerodynamics [41, 19], and deformable object simulations [12, 42]. A popular class of learned neural simulators are Graph Network Simulators (GNSs) [43, 25]. GNSs utilize MPNs, a special type of GNN [44, 23] that representationally encompasses the function class of many classical solvers [33]. GNSs handle physical data by modeling arbitrary entities and their relations as a graph. Notably, all previously mentioned GNSs predict system dynamics iteratively from a given state, whereas we directly estimate entire trajectories, improving rollout stability and prediction speed. Related to our approach is the Equivariant Graph Neural Operator (EGNO) [22], which also predicts full trajectories using SE(3) equivariance to model 3D dynamics and capture spatial and temporal correlations. In this work, we adopt the trajectory prediction framework of EGNOs for our decoder (as shown in Figure 3) but remove the equivariance constraint. We justify this choice in Appendix B.

**Material estimation.** Determining physical parameters from observational data is a challenging and ill-posed problem [26]. Machine learning methods have shown success in inferring material properties from videos [27–29] and point clouds [30], but they rely on knowledge of the underlying

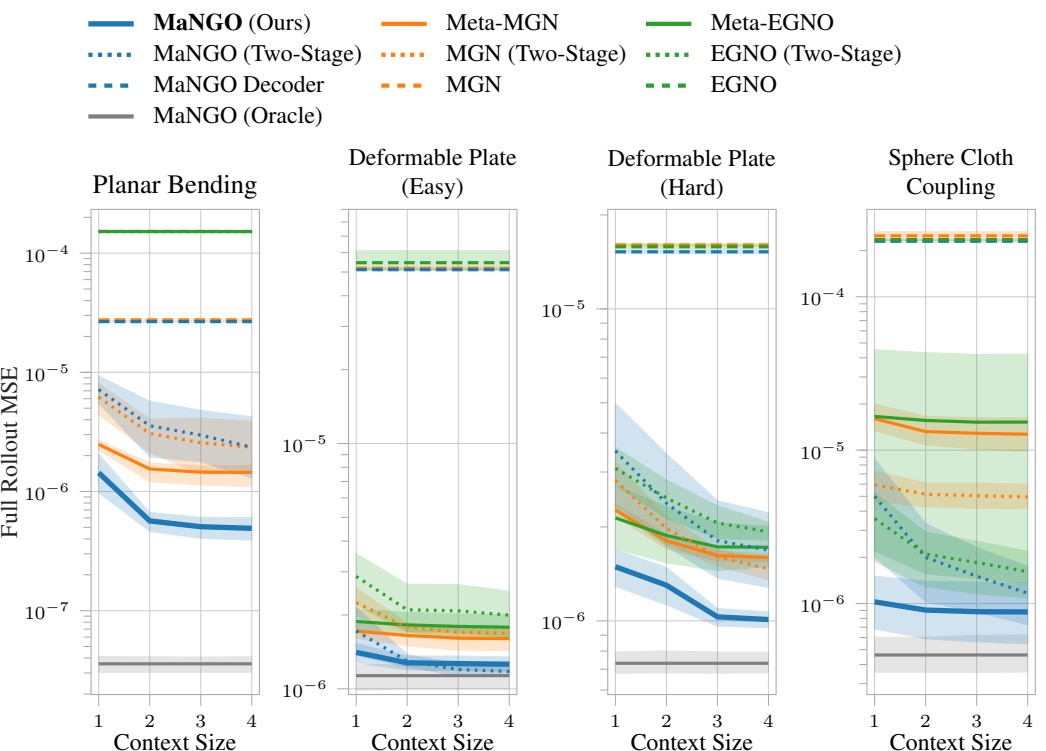

Figure 4: Performance comparison of our proposed methods and baselines across four datasets. We report the mean and 95% confidence interval of the *Full Rollout MSE* over five random seeds. The $x$-axis indicates the number of context samples used by the meta-learning approaches. We compare (i) meta-learning methods that employ a latent simulation parameter representation, (ii) a two-stage setup that explicitly predicts simulation parameters, and (iii) a baseline model that ignores context and performs direct simulation. The MaNGO (Oracle) variant, which has privileged access to the ground-truth simulation parameters, serves as an upper performance bound. Overall, MaNGO consistently outperforms both non–meta-learning and alternative meta-learning approaches, and achieves results close to the oracle.

PDE. Recently, approaches utilizing pre-trained Graph Network Simulators (GNS) to infer material parameters gained popularity due to their computational efficiency and differentiability [31, 32]. By back-propagating through the learned simulator, these methods estimate latent material codes directly from observations. While the approach in Zhao et al. [31] and ours share this goal, only we aggregate a context set of simulation trials to extract the underlying structure. Furthermore, our method does not require any backward pass and model weight updates, resulting in a faster adaptation.

**Meta-learning and Neural Processes.** Learning models that can quickly adapt to small context datasets at test time is often referred to as Meta-Learning. This emerging paradigm has found wide applications in fields such as language models [45] and robotics [46, 47], where it is commonly referred to as In-Context Learning. Meta-Learning is typically categorized into two approaches: optimization-based methods with few-shot examples and context-aggregation methods in a zero-shot fashion. A prominent example of the former is Model-Agnostic Meta-Learning (MAML) [48–51], which employs gradient-based updates to adapt the model to new tasks using few examples. In the latter, Neural Processes (NPs) [34, 52–57] aggregate latent features from a variable-sized context set to produce a latent task description that can be used directly during inference.

In this paper, we adopt Conditional Neural Processes (CNPs) [34] as the core mechanism for aggregating context from different simulation trials to produce a latent description representing the simulation parameters. A GNS is then trained to condition on this latent descriptor, enabling it to adapt dynamically to new materials. To the best of our knowledge, this is the first time such a framework has been proposed, combining meta-learning with graph network simulators to quickly adapt to unknown material properties given only few context-examples without requiring retraining or knowledge of the underlying PDE.

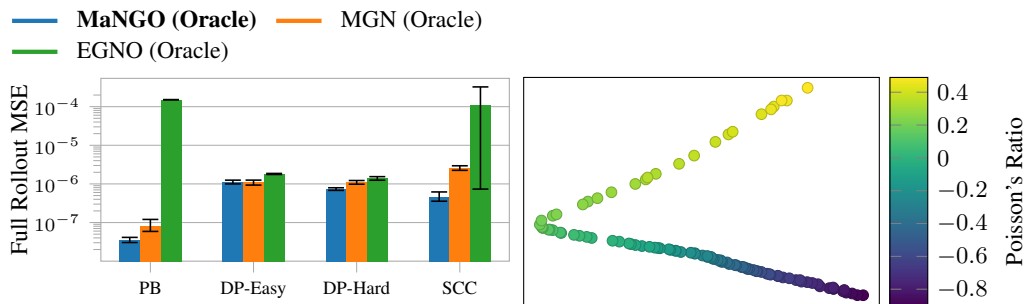

Figure 5: **Left:** Comparison of different GNS decoders with oracle information. MaNGO outperforms both MGN and EGNO, with the performance gap being more visible in the *Sphere Cloth Coupling* task due to its highly complex underlying dynamics. Additionally, EGNO fails to learn in the *Planar Bending* task, a phenomenon further analyzed in Appendix B. **Right:** Visualization of the 2D latent space for Deformable Plate (Easy), with points color-coded by Poisson's ratio. The structured separation (at Poisson value 0) shows that **r** effectively captures the underlying material properties.

## 5 Experiments

We validate MaNGO on four different simulation datasets derived from three distinct simulation platforms. All tasks are normalized to $[0, 1]^3$. The first dataset is a 2D *Deformable Plate* (DP) task [12], which has two variants: *DP-easy* and *DP-hard*. In *DP-easy*, the material property of interest is Poisson's ratio [58]. To increase complexity, *DP-hard* additionally varies Young's modulus and the initial velocity across trials within the same task. Next, we introduce a novel *Planar Bending* (PB) dataset, which simulates the bending of a 2D sheet subjected to two external forces perpendicular to the sheet. In this task, Young's modulus is the property of interest. The final dataset is a new *Sphere Cloth Coupling* (SCC) task, inspired by the Physion++ dataset [59], which involves a coupling system consisting of a sphere and a cloth. In this task, spheres of varying sizes are dropped onto the cloth. Their density is varied to influence the cloth's deformation upon contact.

The length of each simulation trial varies across datasets: 52 time steps for DP, 50 for PB, and 100 time steps for SCC to account for the increased complexity of simulating elastic behaviors. Further details about preprocessing steps and ground-truth simulators can be found in Appendix D.

### 5.1 Training setup and baselines

In this section, we present all methods evaluated in our experiments. Each model variant, whether meta-learning or non-meta-learning, has approximately three million trainable parameters, ensuring a fair comparison across architectures.

**Meta-learning.** We first describe the setup used for our meta-learning framework. Each task dataset $\mathcal{D}_l$ contains 16 different simulation trials, where each trial shares the same material properties $\rho$ (e.g. Poisson's ratio or Young's modulus) but varies in initial conditions (e.g., collider position and size, or force position). During training, we randomly select a subset of $S_l$ of a size between 1 to 8 to serve as the context dataset. This context is then used to predict a random trial[3] using different decoder methods. We compare our proposed MaNGO decoder against the EGNO architecture and a step-based MGN simulator. For testing, we split $\mathcal{D}_*$ into two distinct subsets. The first subset is used as the context dataset, while the second subset provides the initial conditions for predicting the full trajectory. We evaluate various context sizes to assess adaptation capability.

**Two-Stage Training Setup.** Since we assume access to the simulation parameters during training, we compare the meta-learning methods against a two-stage training scheme. In the first stage, we train an encoder to predict the simulation parameters given a simulation trial. In the second stage, we freeze the encoder and train a decoder that uses the predicted simulation parameters from the encoder as an additional input. Therefore, this baseline predicts an explicit representation rather than a latent representation of the simulation parameters. We refer to this approach as MaNGO/MGN/EGNO (Two-Stage).

---

[3]We empirically find that predicting a random trial per batch instead of all trials improves performance.

**Vanilla GNS.** We compare our method against non-meta-learning baselines, including EGNO, and MGN, and our proposed MaNGO decoder without context information. To assess the upper bound, we additionally train an oracle method using the MaNGO decoder which has access to the simulation parameters during training *and test time*. We refer to it as MaNGO-Oracle. For these baselines, the full training set containing all 16 simulation trials is provided. Additionally, for the MGN baseline, we follow the approach of Pfaff et al. [19], adding noise during training to mitigate error accumulation and stabilize rollouts at inference. We tune the input noise level for each task to maximize MGN's performance. The full experimental protocol, along with computational budget details, is available in Section F.

## 5.2 Results

**Main evaluation.** We compare the mean-squared error (MSE) between the ground truth and the predicted simulation averaged over all time steps. Overall, Figure 4 shows that meta-learning approaches consistently outperform non-meta-learning baselines, achieving a relatively low MSE close to the oracle method, MaNGO-Oracle. The two-stage training scheme is only competitive in the simplest environment, Deformable Plate (Easy), and across all environments it requires a larger context size to achieve its best performance. In general, within the meta-learning setting, performance improves with larger context sizes. Furthermore, we emphasize that strong performance is achieved with as few as 2 to 3 context samples (even under the high-complexity DP-hard dataset), highlighting the ability of our meta-learning approach to adapt quickly with minimal context data. All non-meta-learning baselines perform similarly. Interestingly, EGNO struggled to learn in the Planar Bending task, a phenomenon we further analyze in Appendix B. Qualitative visualizations of all methods and tasks are provided in Section G.

To directly compare our proposed MaNGO decoder with existing architectures, we evaluate it in the oracle setting, where material properties are known. As shown in Figure 5 on the left side, MaNGO is outperforming other baselines, confirming the effectiveness of our approach. We suspect that for MGN, auto-regressive prediction suffers from accumulated errors,

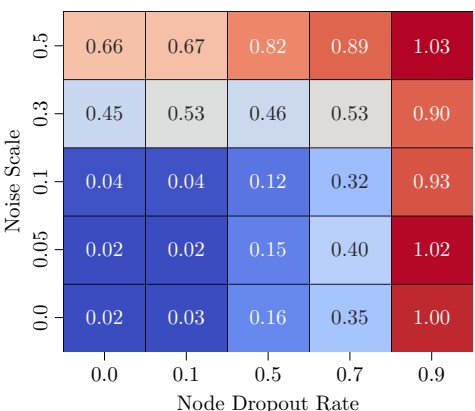

Figure 6: Robustness analysis of MaNGO on the Deformable Plate task under varying levels of Gaussian noise and node dropout, a setup mimicking real-world conditions. The normalized MSE is reported. Our method remains stable with up to $10\%$ noise (relative to the width of the mesh) and $10\%$ node dropout.

an issue particularly evident in the Sphere Cloth Coupling dataset, where highly nonlinear contact dynamics [60, 59] lead to MSE an order of magnitude higher than MaNGO. As for EGNO, while equivariance reduces the amount of required training data, it can also be overly restrictive, as real-world physics is not strictly E(3)-equivariant [61, 62]. Various factors, such as friction and gravity, can break this symmetry, leading to suboptimal generalization.

**Latent visualization.** To understand how the learned latent representation correlates with simulation parameters $\rho$, we visualize the 2D latent space with $\dim(\mathbf{r}) = 2$ for the Deformable Plate task, color-coded by Poisson's ratio. Figure 5 (right) shows a strong correlation between $\mathbf{r}$ and Poisson's ratio, with two linear trends emerging: one from 0 to 0.49 and the other from $-0.9$ to 0. This separation reflects the underlying material behavior — plates with a positive Poisson's ratio expand on contact, while those with a negative Poisson's ratio contract. The learned representation captures this distinction, indicating that the model encodes meaningful physical properties.

**Robustness under sparse and noisy observations.** Inspired by the setup in [31], we evaluate the robustness of MaNGO on the Deformable Plate task by introducing Gaussian noise into the observational context data and reducing the number of observed nodes at test time. To this end, we also introduce a small noise level of 0.05 for the context split during training to enhance the robustness of the encoder. We report the normalized MSE in the range $[0, 1]$, where the minimum is set by MaNGO-Oracle and the maximum by the non-meta-learning MaNGO approach (as shown in

Figure 4). As shown in Figure 6, the lower-left region of the matrix—corresponding to a 10% noise level and 10% node dropout—demonstrates near-optimal performance. Even with 50% of nodes unobserved, the performance drop remains around 15%, highlighting the model's ability to handle sparse and noisy observations. These results confirm that our encoder is robust under such conditions, reflecting real-world scenarios.

**Runtime Efficiency and Memory Consumption** A key advantage of MaNGO is its ability to predict entire trajectories in a single forward pass, enabling efficient batched inference over time. This design leads to substantial improvements in inference speed compared to traditional autoregressive next-step simulators. On the Sphere-Cloth Coupling benchmark, the CNP encoder requires approximately 6 ms to compute the latent representation, and the MaNGO decoder simulates the full trajectory in only 13 ms. In contrast, the MGN next-step simulator takes about 500 ms, as it predicts one step at a time, making MaNGO over an order of magnitude faster than the already efficient autoregressive neural simulator.

However, the batched inference and training scheme comes with increased memory requirements. GPU memory consumption scales approximately linearly with the number of predicted time steps, since the model retains intermediate activations across the temporal dimension. In our experiments, we observe that

$$\text{Memory(MaNGO)} \approx \text{Memory(MGN)} \times \text{Number of predicted steps.}$$

## 6    Conclusion

In this work, we explored data-driven adaptation of graph network simulators (GNS) via meta-learning. Specifically, we investigated the setting where simulation parameters are unknown at test time which would require retraining or labor-intensive data collection for existing methods. In a series of experiments, we demonstrated the potential of meta-learning for GNS, where our approach achieves accuracy on unseen material properties comparable to that of an oracle model. We view this work as a first stepping-stone towards the next generation of data-driven simulators, that are fast, differentiable, and capable of adapting to a wide variety of simulation settings. A discussion of the broader impact of this work is provided in Section A.

**Limitations and Future Work**   One limitation of our current approach is its focus on a single data modality at test time. In practical scenarios, however, test-time data may come in diverse formats, such as point clouds captured by cameras. Our method does not yet support such modalities, which constrains its applicability in real-world settings. Addressing this limitation would require the development of new architectures capable of handling heterogeneous data.

A further limitation is that MaNGO assumes a fixed graph topology and requires substantial GPU memory when predicting full trajectories in a single forward pass. These constraints limit its scalability and applicability to systems with dynamic connectivity. A promising direction for future work is to adopt a hybrid decoding strategy that predicts shorter temporal segments while dynamically updating the graph structure after each segment. Such an approach could reduce memory demands and enable the modeling of systems with evolving topologies, bridging the gap between efficiency and flexibility.

## Acknowledgements

This work is part of the DFG AI Resarch Unit 5339 regarding the combination of physics-based simulation with AI-based methodologies for the fast maturation of manufacturing processes. The financial support by German Research Foundation (DFG, Deutsche Forschungsgemeinschaft) is gratefully acknowledged. The authors acknowledge support by the state of Baden-Württemberg through bwHPC, as well as the HoreKa supercomputer funded by the Ministry of Science, Research and the Arts Baden-Württemberg and by the German Federal Ministry of Education and Research. We thank Philipp Becker for suggesting the title acronym used in this work.

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

## A    Braoder Impact Statement

Our proposed Graph Network Simulator, with its ability to adapt to new material properties through a small context set, offers significant advancements across fields that rely on computational modeling and simulation. By reducing the need for extensive simulation data and recalibration, it can lower computational costs while maintaining high accuracy. This adaptability could benefit industries ranging from materials science to robotics, enabling more efficient design and testing of novel materials.

However, this flexibility in simulating a wide range of material properties could also be misused in contexts where precise material behavior is critical, such as in the development of advanced weaponry or other high-risk technologies. While the primary intent is to advance scientific and engineering applications, ethical considerations must be taken into account to prevent unintended harmful uses.

## B    Limitations of the Equivariant Graph Neural Network

In this section, we analyze the limitations of the Equivariant Graph Neural Network (EGNN) [63], which serves as the GNN backbone of the Equivariant Graph Neural Operator (EGNO) baseline [22]. During our experiments, we observed that EGNO fails to predict any deformation in the Planar Bending task.

In this task, all neural operators receive an initial input configuration $\mathbf{p}_0$, representing a completely flat plane as the positions of the mesh. Consequently, all points in this initial graph lie within a plane $E$ embedded in three-dimensional space. In this appendix, we demonstrate that the spatial output of any trained EGNN will also remain confined to a plane. This implies that an EGNN cannot solve the Planar Bending task, as the required deformed state does not lie within a single plane.

We establish this result by first proving that an EGNN maps all points in the specific plane

$$\mathbf{Z}_{=0} = \{(x, y, z) \in \mathbb{R}^3 \mid z = 0\}$$

back onto the same plane $\mathbf{Z}_{=0}$. To see this, consider any node $v \in \mathcal{G}$ with an initial position $\mathbf{p}_v^0 = (x, y, 0) \in \mathbf{Z}_{=0}$. Applying a single layer of EGNN to update the node positions, we obtain

$$\mathbf{p}_v^1 = \mathbf{p}_v^0 + C \sum_{w \in \mathcal{N}(v)} (\mathbf{p}_v^0 - \mathbf{p}_w^0)\varphi_x(\mathbf{m}_{vw}).$$

Examining the $z$-coordinate, we note that the message term $\varphi_x(\mathbf{m}_{vw})$ is scaled by $(\mathbf{p}_v^0 - \mathbf{p}_w^0)$. Since all $\mathbf{p}_v^0$ lie in $\mathbf{Z}_{=0}$, their $z$-coordinates are zero, meaning $(\mathbf{p}_v^0 - \mathbf{p}_w^0)$ has no $z$-component. Consequently, the updated $z$-coordinate in $\mathbf{p}_v^1$ remains zero, ensuring that $\mathbf{p}_v^1 \in \mathbf{Z}_{=0}$. By induction, this property holds for all subsequent layers, proving that EGNN maps $\mathbf{Z}_{=0}$ onto itself.

Next, we extend this result to any arbitrary plane $E$ in three-dimensional space. Since $E$ is a plane, there exists a transformation $T \in \mathrm{SE}(3)$ such that

$$T(E) = \mathbf{Z}_{=0}.$$

By the equivariance property of EGNN, there exists another transformation $S \in \mathrm{SE}(3)$ satisfying

$$\mathrm{EGNN}(\underbrace{T(\mathcal{G})}_{\subset \mathbf{Z}_{=0}}) = S(\mathrm{EGNN}(\mathcal{G})).$$

From our previous result, the left-hand side remains confined to $\mathbf{Z}_{=0}$. Thus, applying $S^{-1}$ yields

$$\mathrm{EGNN}(\mathcal{G}) \subset S^{-1}\mathbf{Z}_{=0},$$

which is another plane in three-dimensional space. This establishes the key result: an EGNN always maps planar inputs to planar outputs, rendering it incapable of solving the Planar Bending task.

A similar argument extends to the full EGNO architecture since its equivariant temporal convolution layer in Fourier space is also equivariant and it preserves the property of mapping $\mathbf{Z}_{=0}$ onto itself. Thus, EGNO, like EGNN, is inherently unable to capture the required deformations in the Planar Bending task.

## C   Feature creation for the MaNGO decoder

This appendix provides a detailed explanation of the input processing for the MaNGO simulator. First, similar to the Spatiotemporal encoder, we create a sequence of graphs $\mathcal{G}_t$ over time. However, unlike the encoder, we repeat the initial positions and velocities of all deformable nodes across time, as the future positions—our target variable—are unknown and need to be estimated. In $\mathcal{G}_t$, we define the node features for a node $v$ as the tensor $[\mathbf{r}, \mathbf{h}_v, \mathbf{v}_t, \mathrm{TE}(t)]$ consisting of the latent system identification $\mathbf{r}$, the node features $\mathbf{h}_v$, the velocity $\mathbf{v}_t$ and a time embedding of the time step $t$. Note that the position is not part of the node features. Instead, following [19], we compute for each edge $e$ the relative position $\mathbf{p}_{e,t}^{\mathrm{rel}}$ to ensure translation invariance. The complete edge features are given as $[\mathbf{e}_e, \mathbf{p}_{e,t}^{\mathrm{rel}}, \mathrm{TE}(t)]$ consisting of the (time-independent) edge features, the relative position and a time step embedding.

## D   Datasets and Preprocessing information

In this section, we provide detailed information about the datasets used in our experiments. Table 1 summarizes the key characteristics of each dataset, including the dataset splits, simulation length, and the number of nodes used for prediction.

For brevity, we use the following abbreviations throughout the paper:

- **PB**: *Planar Bending*.
- **DP**: *Deformable Plate*, with two variants: *DP-easy* and *DP-hard*.
- **SCC**: *Sphere Cloth Coupling*.

Table 2 further details the training setup for each dataset, specifying the material properties considered and the variations in initial conditions. These variations influence the dynamics of each task, ensuring diverse training scenarios that test the generalization capabilities of our method.

Table 1: Dataset descriptions

| Name | Train/Val/Test Splits | Number of Steps | Number of Nodes for Prediction |
|------|----------------------|-----------------|-------------------------------|
| PB | 460/50/50 | 50 | 225 |
| DP-easy | 600/100/100 | 52 | 81 |
| DP-hard | 600/100/100 | 52 | 81 |
| SCC | 600/100/100 | 100 | 400 (cloth) + 98 (sphere) |

Table 2: Training setup for each dataset

| Name | Material Properties | Initial Condition Variations |
|------|--------------------|-----------------------------|
| PB | Young's modulus | Two forces: $(x, y)$ position, $\{-1, 1\}$ direction, constant magnitude |
| DP-easy | Poisson's ratio | Collider's $x$ position, size, constant initial velocity |
| DP-hard | Young's modulus, Poisson's ratio | Collider's $x$ position, size, **varied** initial velocity |
| SCC | Sphere's density | Sphere's size, same initial position |

**Planar Bending.** We uniformly select Young's modulus from 10 to 500, from a very deformable to an almost stiff sheet. The boundary nodes of the sheet are kept in place.

**Deformable Plate.** The original task was introduced in [12], generated using Simulation Open Framework Architecture (SOFA) [64]. We extended to meta-learning setting by sampling Poisson's ratios between $-0.9$ to $0.49$, under different trapezoidal meshes. We further increase the difficulty of this dataset by also randomizing the Young's modulus within a range from 500 to 10000 using Log-Uniform distribution.

**Sphere Cloth Coupling.** Each trajectory in this dataset is generated by selecting a sphere radius from the range $[0.2, 0.8]$. The material property of interest is the sphere's density, which varies between $[2.0, 100.0]$. The cloth is initialized in a stable state, remaining consistent across all tasks and trials. This dataset is created using NVIDIA Isaac Sim [65], which leverages PhysX 5.0 [66] to simulate position-based dynamics (PBD) particle interactions.

Table 3: **Left:** Training setup for each dataset. **Right:** Noise-scale per task for Auto-regressive methods

| Parameter | Value |
|---|---|
| Node feature dimension | 128 |
| Latent representation dimension | 128 |
| Decoder hidden dimension | 128 |
| Message passing blocks | 15 |
| GNN Aggregation function | Mean |
| GNN Activation function | Leaky ReLU |
| Learning rate (Auto-regressive methods) | $5.0 \times 10^{-4}$ |
| Learning rate | $1.0 \times 10^{-4}$ |
| Optimizer | AdamW [67] |
| Min Context Size (Training) | 1 |
| Max Context Size (Training) | 8 |
| MaNGO-CNN Decoder Kernel size | 7 |
| CNN-Deepset Encoder Kernel size | 3 |
| Latent representation aggregation | Maximum |

| Task | Value |
|---|---|
| PB | $5.0 \times 10^{-4}$ |
| DP-easy | $7.0 \times 10^{-4}$ |
| DP-hard | $7.0 \times 10^{-4}$ |
| SCC | $1.0 \times 10^{-3}$ |

For meta-learning setup, we assume a set $S_i$ containing 16 different simulation trials, where each trial shares the same material properties (e.g., density, Poisson's ratio, Young's modulus) but varies in initial conditions (e.g., sphere size, force positions, and magnitudes).

# E    MGN Decoder for Conditional Neural Processes

In step-based prediction tasks, batches are typically shuffled to include different simulations and time steps. However, when using a Conditional Neural Process (CNP), only data from the same task can be used for each batch, which can impact performance. During hyper parameter optimization, we tested a modified version of the MGN where we reduced batch shuffling to mimic the CNP approach. This modification resulted in poorer performance compared to the standard MGN. This difference in performance may partially explain why Meta-MGN underperforms relative to MaNGO.

# F    Experimental Protocol

In order to promote reproducibility, we provide details of our experimental methodology. Table 2 presents the hyperparameters used in our experiments. For a comprehensive description of the creation of all datasets, please refer to Appendix D.

The training took place on an NVIDIA A100 GPU, with each method given the same computation budget of 48 hours. Consequently, the number of epochs varied, as the batching differed significantly between the meta-learning methods and the step-based MeshGraphNet (MGN). In total, generating the results presented in this paper required approximately 8,500 GPU hours.

We conducted a multi-staged grid-based hyperparameter search for the learning rate, input noise, and other hyperparameters as residual connections and layer norms. We did not use the test data for this, but tuned all hyperparameters on a separate validation split. This split was also used to determine the best epoch checkpoint to mitigate any overfitting effects. Hyperparameter tuning required an additional computational budget of approximately 6,000 GPU hours.

For MGN, we included velocity features of the current step.

# G    Visualizations

In this section, we present qualitative results for all tasks and methods discussed in the main paper.

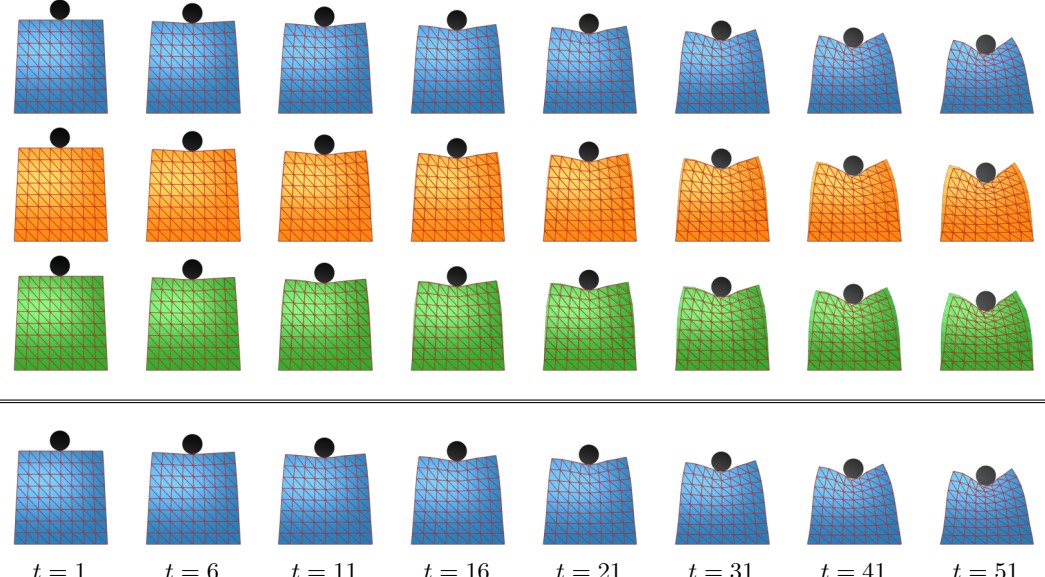

$t = 1$     $t = 6$     $t = 11$     $t = 16$     $t = 21$     $t = 31$     $t = 41$     $t = 51$

Figure 7: Simulation over time of an exemplary test trajectory from the **DP-easy** task. The figure compares predictions from MaNGO, MGN, and EGNO. The last row, MaNGO-Oracle, is separated by a horizontal line and represents predictions using oracle information. The **context set size** is set to $4$. All visualizations show the colored **predicted mesh**, with a **wireframe** representing the ground-truth simulation. MaNGO accurately predicts the correct material properties, leading to a highly accurate simulation.

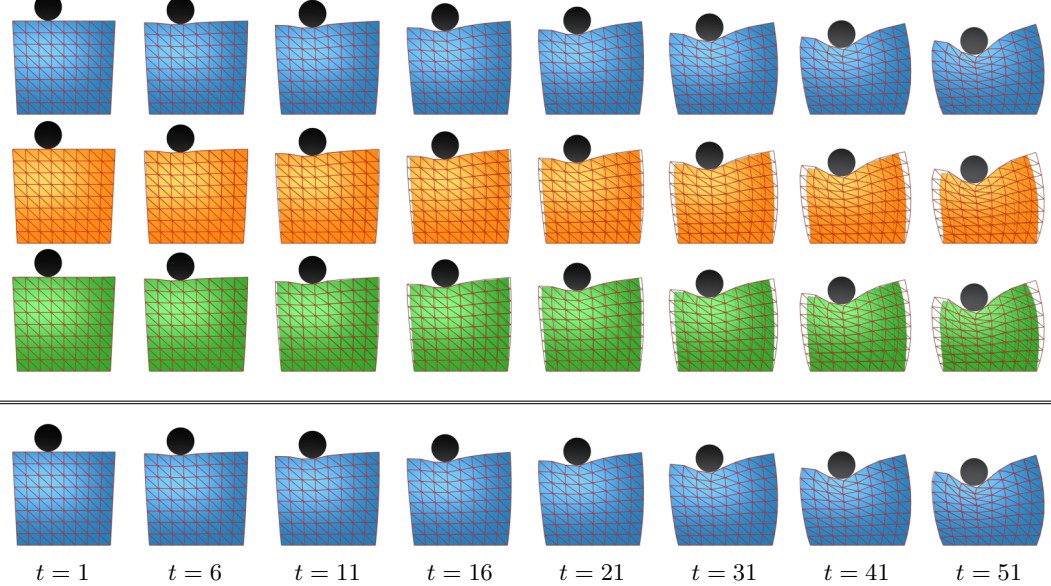

$t = 1$     $t = 6$     $t = 11$     $t = 16$     $t = 21$     $t = 31$     $t = 41$     $t = 51$

Figure 8: Simulation over time of an exemplary test trajectory from the **DP-hard** task. The figure compares predictions from MaNGO, MGN, and EGNO. The last row, MaNGO-Oracle, is separated by a horizontal line and represents predictions using oracle information. The **context set size** is set to $4$. All visualizations show the colored **predicted mesh**, with a **wireframe** representing the ground-truth simulation. MaNGO accurately predicts the correct material properties, leading to a highly accurate simulation.

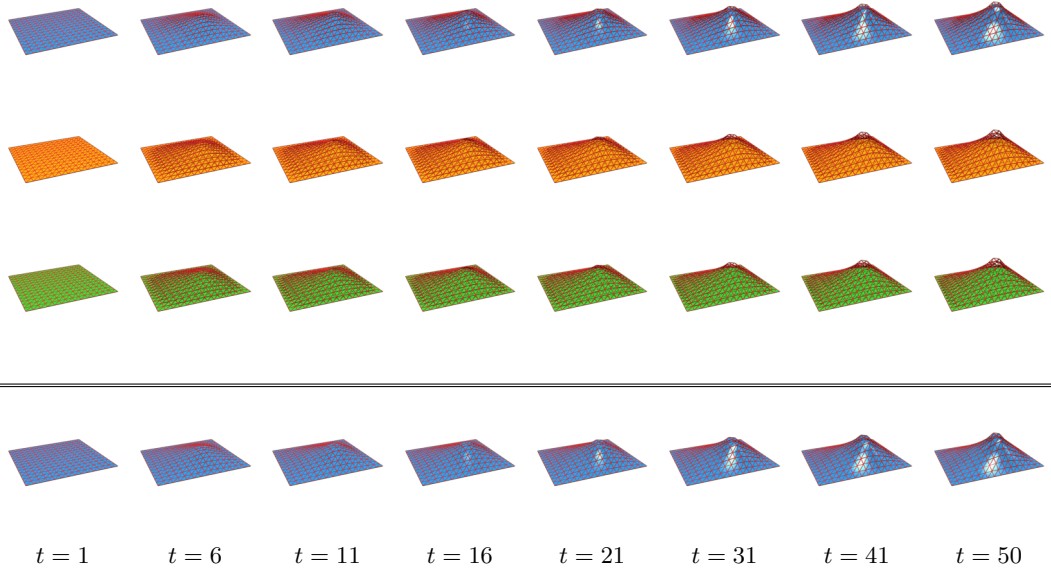

$t = 1$  $t = 6$  $t = 11$  $t = 16$  $t = 21$  $t = 31$  $t = 41$  $t = 50$

Figure 9: Simulation over time of an exemplary test trajectory from the **PB-easy** task. The figure compares predictions from **MaNGO**, MGN, and EGNO. The last row, MaNGO-Oracle, is separated by a horizontal line and represents predictions using oracle information. The **context set size** is set to 4. All visualizations show the colored **predicted mesh**, with a **wireframe** representing the ground-truth simulation. **MaNGO** accurately predicts the correct material properties, leading to a highly accurate simulation.

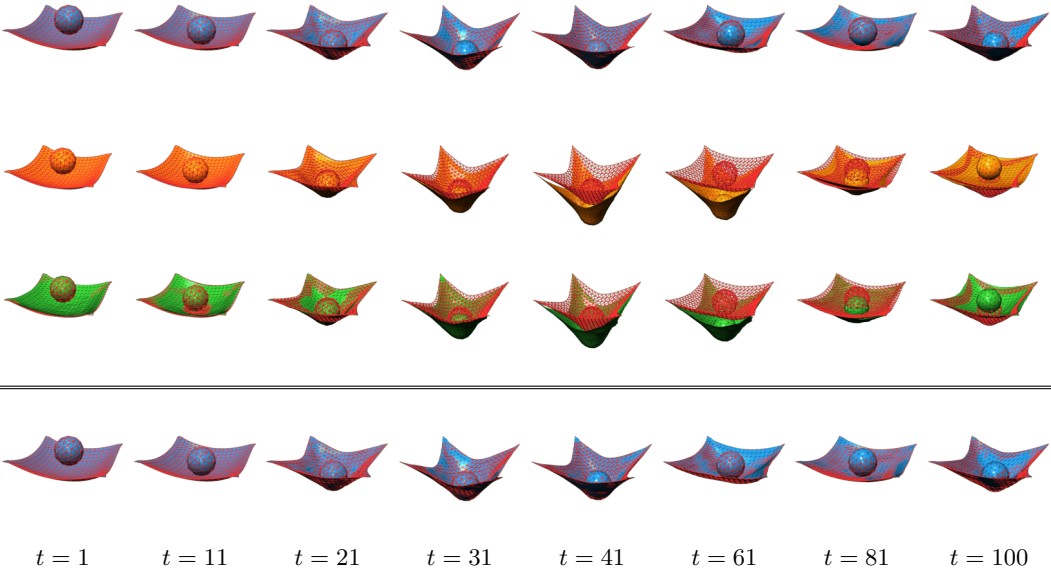

$t = 1$  $t = 11$  $t = 21$  $t = 31$  $t = 41$  $t = 61$  $t = 81$  $t = 100$

Figure 10: Simulation over time of an exemplary test trajectory from the **Sphere Cloth Coupling** task. The figure compares predictions from **MaNGO**, MGN, and EGNO. The last row, MaNGO-Oracle, is separated by a horizontal line and represents predictions using oracle information. The **context set size** is set to 4. All visualizations show the colored **predicted mesh**, with a **wireframe** representing the ground-truth simulation. **MaNGO** accurately predicts the correct material properties, leading to a highly accurate simulation.

