# OpenReview forum: "MaNGO — Adaptable Graph Network Simulators via Meta-Learning"
_NeurIPS.cc/2025/Conference — NeurIPS 2025 poster_

### Official Review · Reviewer_EtEt · 2025-06-10

**Clarity:** 2
**Significance:** 2
**Originality:** 3
**Rating:** 4
**Confidence:** 4

**Summary:**

__I have reviewed this paper for ICML. As far as I can tell, the only change to the paper is the title. I will hence re-post my review from ICML.__

The paper introduces a new GNN-based method for learning physical dynamics. It specifically focusses on latent parameter estimation. Given a set of reference simulations, an encoder create a latent embedding r, which contains an estimate of hidden simulation parameters. A GNN decoder can then use r to perform inference with those material parameters.

**Questions:**

(my questions were already answered in the last review process)

**Ethical Concerns:**

["NO or VERY MINOR ethics concerns only"]

**Final Justification:**

This is a borderline paper-- There's something there, and the method is interesting, but to be really impactful the paper would need more realistic experiments (ideally real-world data or at least a complex material model to really justify the method) and better ablations of the method.
That said, adding the two-stage experiment to the paper is already a good step. I revised my score from 'borderline reject' to 'borderline accept'.

**Limitations:**

yes, except for what is mentioned above.

**Paper Formatting Concerns:**

---

**Quality:**

2

**Strengths And Weaknesses:**

The paper claims better performance on physical simulation by a) a new GNN architecture and b) meta-learned latent parameters (CNP). The claims are supported by experiments, although experimental validation could be improved.

The method is quite complex, with many choice different to previous work, and it's not immediately obvious to me if they are actually needed or effective.
- A lot of the complexity comes from CNP, i.e. inferring a latent representation of the hidden material parameters. But then, 3.2. introduces a direct regression loss against the true latent parameters. But if rho has to be assumed known in training, there are much simpler options which need to be tested. The most obvious is training an encoder from D_L -> rho, and conditioning the decoder on rho instead of r. This is much simpler: no Gaussian distribution needed, no potential issues with irrelevant information contaminating the latent. Plus encoder and decoder can be trained separately, severely reducing memory and training time. I would imagine this would actually perform quite well for the datasets investigated in this paper, given the strong performance of the oracle models. (there may be some downsides for ambiguous real-world data, but that's not studied here)
- The architecture of the encoder is also a bit surprising to me. For (traditional) estimation of material parameters, you'd need spatial and temporal derivatives. Deep sets without any neighborhood information will struggle computing spatial gradients; something like 1 or 2 message passing steps with a short temporal window, and then aggressive spatiotemporal pooling seems like a more obvious choice.
- EGNO works because the temporal convolution is in Fourier space, i.e. a global operation. Replacing this with a 1D CNN means information will only travel locally, which will likely limit the ability to model long sequences or sequences with complex dynamics (all examples are quite short and contain simple motion).

There are a lot of novel surprising choices in the model. These need to be properly ablated, and compared to simpler choices (see e.g. the points mentioned under Methods And Evaluation Criteria). I feel the paper tries to do too much, without spending the time to properly evaluate all of its contributions, and we don't end up learned all that much in the end.

For example, there's actually an interesting point in the core GNN architecture of MGN vs MGNO vs EGNO. EGNO isn't quite made for the type of system studied here, but there may be advantages of this style of approach compared to autoregressive MGN. But this needs to be properly studied, with detailed analysis on pros and cons (e.g. ingesting entire trajectories quite obviously comes with much higher memory footprint, and it's probably not a coincidence that the systems studied here are extremely low-dimensional), ablations to figure out the best configuration for different system types and sizes, etc. There might be good tradeoffs to be had, but this really needs to be its own paper.

Similarly, material parameter estimation is an important topic for using learned simulator in real-world settings. For this topic, the focus really should be on how to best design an encoder, how to deal with uncertainty (and when that is/isn't important), latent vs. explicit parameter estimation, etc. And most importantly, parameter estimation is fundamentally about real world data (for simulations we can just assume parameters are available). So studying real-world data, or at least a good proxy (where training and testing material model/parameters have a gap) is probably a good idea.

---

> ### Author Rebuttal · Authors · 2025-07-30
>
> We thank the reviewer for their time and for re-engaging with our submission. We appreciate the opportunity to respond to the points raised and provide clarification where needed. While we understand that parts of the submission may be familiar, we respectfully acknowledge that each venue offers a fresh review process, and we are grateful for the paper being considered within the context of NeurIPS.
>
> ### **On End-to-End Latent Inference vs. Two-Stage Regression**
>
> The primary motivation behind our approach is the **joint shaping of the latent representation** through **end-to-end training**. While we acknowledge that a two-stage pipeline (e.g., directly regressing material parameters and conditioning on them) may work well on simpler problems, we argue that our method promotes a more **expressive and integrated latent space**. This space captures not only the identity of the physical parameters but also their **functional impact on the system’s dynamics**.
>
> This is exemplified in **Figure 5b**, where the learned latent representations exhibit a clear and meaningful separation—for example, between **positive and negative Poisson’s ratios**, corresponding to expanding and contracting materials. This reflects the latent’s ability to encode **task-relevant structure**, rather than merely regressing known quantities.
>
> ### **On the Use of Deep Sets Instead of Message Passing in the Encoder**
>
> We appreciate the reviewer’s suggestion and agree that message passing is an intuitive choice for many graph-based problems. In fact, we implemented and tested a **message-passing-based encoder** early in development. However, we found **no significant improvement** over the simpler and more efficient **Deep Sets-based encoder**.
>
> We hypothesize that this is due to the **global nature** of the material parameters being inferred. Since the encoder has access to **node positions**, Deep Sets can effectively represent the necessary global characteristics. In contrast, message passing is particularly suited for modeling **localized interactions**, which are less relevant when estimating global latent variables such as material properties.
>
> ### **On the Finite Receptive Field of Temporal Convolutions**
>
> We acknowledge the concern regarding the **limited receptive field** of 1D CNNs, especially compared to EGNO's use of global Fourier-based convolutions. However, our design intentionally leverages **local temporal modeling**, which we find more effective for capturing **short-horizon dynamics**, particularly in the presence of **contact interactions** and **localized temporal events**.
>
> Unlike EGNO, our model operates directly in the **time domain**, enabling it to learn **localized temporal dependencies** without assuming global periodic structure. In all experiments, we ensure that the CNN’s **receptive field fully covers the sequence length** used for training and evaluation, mitigating the risk of incomplete temporal context.
>
> ### **On Memory Usage and Trajectory Length**
>
> We agree that ingesting full trajectories introduces a higher **memory footprint**, which is a known trade-off of non-autoregressive models. In MaNGO, memory usage scales **linearly with the number of predicted time steps**, as the network retains activations across the full temporal sequence:
>
> Memory(MaNGO)≈Memory(MGN)×Number of predicted steps
>
> That said, this design enables **significant inference speed-ups** by eliminating the need for sequential rollout.
>
> On the **Sphere-Cloth Coupling** benchmark:
>
> - The **CNP encoder** takes approximately **6 ms** to compute the latent representation rrr,
> - The **MaNGO decoder** predicts the full trajectory in **~13 ms**,
> - By comparison, the **autoregressive MGN** requires **~500 ms**, as it generates one step at a time.
>
> To address the memory challenge, we propose a **hybrid decoding strategy** as a promising extension. Instead of predicting the full sequence at once, the model could generate **smaller temporal windows** (e.g., 10–20 steps) in sequence. This would substantially reduce peak memory usage while maintaining the benefits of batched inference.
>
> We appreciate the reviewer’s insights and believe the above clarifications help explain the reasoning and trade-offs behind our architectural choices. We hope this response provides useful context for evaluating the submission.

---

> > ### Comment · Reviewer_EtEt · 2025-08-05
> >
> > Thank you for the clarifications.
> >
> > Overall, this is not a bad paper, and I'm not opposed to seeing it published. I just think it could be a much more impactful by better evaluating (and maybe revising) its methods.
> >
> > As pointed out in my review, I think the biggest issue is that the setup is quite artificial. I actually agree that a simple, dumb 2-stage approach (first estimate material parameters, and then run a plain model conditioned on these parameters) will likely fails for complex material parameter spaces, or real-world problems. But in the simple experiments shown in the paper, I think it's quite likely to do well. So I really think that would be a necessary baseline to test against; and also ideally we'd see more complex material parametrizations. I think we'd learn a lot by being able to observe under which the simple approach breaks done, and where the paper's more complex approach might become necessary.
> >
> > Did you have the chance to perform any more experiments on this front, or on MGNO/EGNO ?
> > Also, if there is any changes in the paper compared to the rebuttal-version of ICML please let me know, as I may have missed it.

---

> > > ### Author Response · Authors · 2025-08-07
> > >
> > > We thank the reviewer for the continued discussion. In this version, we focused mostly on improving the writing and presentation. We made the **story** a bit **more streamlined** and concise and updated the figures for a uniform layout. Beyond these presentational changes, the core technical contribution of the paper remains largely unchanged.
> > >
> > >
> > > That being said, we did **evaluate the suggested 2-stage setup**. We found that the results are largely comparable to either Meta-MGN or MaNGO on the considered experiments, depending on the decoder used. In particular, the **MaNGO** decoder, which predicts the full remaining trajectory, **still outperforms a naive 2-stage MGN-like approach, showing that the MaNGO decoder is a crucial contribution on its own**. We also do not believe that a 2 stage approach is simpler. In terms of architecture, it requires the same complexity (context aggregation of time series for decoding the parameters and a MaNGO decoder for the prediction) but now we need to solve 2 learning problems where we potentially have to optimize  hyper-parameters (learning rates, weight decay, etc..) individually to avoid overfitting. Additionally, the first stage of training requires some sort of validation set to prevent overfitting.  In contrast, our approach is optimized end-2-end. Furthermore, it could be trained in semi-supervised scenarios, where only a subset of the data contains the labels for the physics parameters. Similarly, our architecture could potentially be fine-tuned with data from real-systems, where the physics parameters are unknown. This is not feasable with a 2 stage approach.  We also agree with the reviewer’s assessment that our end-to-end training should yield advantages on more complex tasks. While those points are out of scope for the current submission, they present an interesting and promising avenue for future work.
> > >
> > >
> > > We thus omitted it in this version of the paper for the sake of a simpler, more streamlined story and more concise writing. However, we take the reviewer's point that this baseline **provides a more transparent and holistic comparison** and will include it in the final revision.

---

> > > > ### Comment · Reviewer_EtEt · 2025-08-08
> > > >
> > > > Thank you for the response. I think including this additional study will definitely strengthen the paper, so I would encourage the authors to include it in the final revision if the paper is accepted.
> > > >
> > > > I have now updated my review & revised my score.

---

### Official Review · Reviewer_wdos · 2025-06-30

**Clarity:** 3
**Significance:** 3
**Originality:** 2
**Rating:** 4
**Confidence:** 3

**Summary:**

This paper introduces MaNGO, a meta-learning framework for graph-based physics simulators. The goal is to enable fast adaptation to new, unseen physical parameters (e.g., material properties) without retraining. MaNGO uses Conditional Neural Processes (CNPs) to infer a latent representation of these parameters from a few "context" examples. This representation then conditions a novel, non-autoregressive decoder—which combines Message Passing Networks (MPNs) for spatial dynamics and a 1D CNN for temporal dynamics—to predict the full simulation trajectory.

**Questions:**

1.The paper notes that the neural operator framework is more memory-intensive than autoregressive methods. Could you elaborate on the memory complexity and scalability of the non-autoregressive MaNGO decoder, particularly when dealing with long time-series and large-scale graphs ?
2.Is there a practical limit or "tipping point" where MaNGO's memory consumption becomes a significant drawback compared to autoregressive models?
3.The Deep Set encoder aggregate features from samples into a latent representation. Have you explored alternative, potentially more expressive, aggregation mechanisms?  Was there a comparative study to justify the choice of Deep Sets?
4.Could you please elaborate on the experimental setup for data partitioning? Specifically, how were the tasks divided into meta-training and meta-test sets? Whether to test samples that have never been seen before during the test？

**Ethical Concerns:**

["NO or VERY MINOR ethics concerns only"]

**Final Justification:**

I thank the authors for their detailed and thoughtful rebuttal. While the rebuttal provided valuable clarifications and addressed several specific points, I am maintaining my original "Borderline Accept" (Rating 4) recommendation for the reasons outlined below.
Key Issues Resolved (and how):
Autoregressive Comparison (Weakness 1): The authors clarified that Meta-MGN is indeed included in the comparisons (e.g., Figure 4), and the provided quantitative runtime analysis (MaNGO Decoder ~13ms vs. Meta-MGN ~500ms) strongly supports the efficiency claims. This issue is now sufficiently clarified.
Encoder Choice (Weakness 3 / Question 3): The authors provided a reasonable justification for selecting the Deep Set encoder, citing their own experimentation and hypothesizing that its global aggregation is sufficient for inferring global material parameters.
Memory Complexity & Scalability (Questions 1 & 2): My questions regarding MaNGO's memory complexity, scalability, and practical limits were clearly addressed. The explanation of linear memory scaling and the proposal of a hybrid decoding strategy for long trajectories are helpful and provide a path forward for large-scale applications.
Experimental Setup & Data Partitioning (Question 4): The detailed clarification of the meta-learning data partitioning protocol is highly appreciated and confirms the rigor of the experimental setup, ensuring no data leakage from meta-training to meta-testing.
Key Issues Remaining Unresolved (and why they are weighted):
Limited Meta-Learning Baselines (Weakness 2): This crucial concern was not addressed in the rebuttal. The paper lacks a comparison with other prominent meta-learning approaches, such as optimization-based methods like MAML. This omission is significant as it prevents a comprehensive understanding of MaNGO's performance and superiority within the broader meta-learning landscape. Without this, the claims about the proposed CNP-based framework's overall superiority in the meta-learning context are not fully substantiated.
Weight Assigned:
While I acknowledge and appreciate the authors' thorough responses to the majority of my questions and weaknesses, the absence of a comparison with other major meta-learning paradigms (Weakness 2) remains a substantial gap. This particular limitation directly impacts the paper's claimed generalizability and its positioning within the meta-learning field. The strengths of the paper (important problem, innovative decoder, clarity) are notable, and the resolved issues certainly clarify aspects of the proposed method. However, the unaddressed methodological comparison prevents a stronger endorsement beyond a borderline acceptance, as it limits the scope of the empirical validation.
Recommendation: Borderline Accept.

**Limitations:**

no,I think some comparative experiments should be added.

**Paper Formatting Concerns:**

I didn't notice the format issue.

**Quality:**

3

**Strengths And Weaknesses:**

Strength
1.Important Problem: The paper tackles the critical and practical problem of simulator adaptation, which is a major bottleneck in data-driven science and engineering.
2.Innovative Decoder Design: The non-autoregressive decoder, blending the strengths of MPNs and 1D CNNs, is a smart solution that effectively avoids common issues like error accumulation (from autoregression) and over-constraining (from strict equivariance).
3.Clarity and Organization: The paper is well-written and logically structured. The motivation, methodology are presented clearly, making the paper easy to follow.
Weakness
1.Missing Comparison with Autoregressive Models: The paper claims its non-autoregressive decoder is more memory-intensive but faster than autoregressive ones. However, it lacks a direct experimental comparison with an autoregressive baseline (like a meta-learning version of MGN) to substantiate this trade-off..
2.Limited Meta-Learning Baselines: The superiority of the proposed CNP-based framework is not fully established, as it is not compared against other prominent meta-learning approaches, such as optimization-based methods like MAML.
3.The encoder uses a Deep Set to aggregate context features. This might be an oversimplified approach, and the paper does not explore or justify this choice over potentially more sophisticated aggregation methods.

---

> ### Author Rebuttal · Authors · 2025-07-30
>
> We thank the reviewer for their detailed and thoughtful feedback. Below, we address each of the raised points:
>
> ### **Missing Comparison with Autoregressive Models**
>
> We would like to clarify that we **do include a direct comparison with a meta-learning version of MGN (Meta-MGN)** in our experiments. This baseline is shown in **orange** in the plots (e.g., **Figure 4**), and is discussed throughout the experimental section.
>
> - MaNGO consistently **outperforms Meta-MGN both quantitatively and qualitatively** across all benchmark tasks. Visual comparisons can also be found in the **appendix visualizations**.
> - In terms of **runtime**, MaNGO demonstrates significant efficiency due to its non-autoregressive architecture. Comparing on the Sphere Cloth Coupling task, we get:
>     - **CNP Encoder**: ~6 ms
>     - **MaNGO Decoder** (full trajectory): ~13 ms
>     - **Meta-MGN (autoregressive)**: ~500 ms
>
>         All models were trained for the same number of steps to ensure a **fair comparison**.
>
>
> These results support our claim that MaNGO offers a favorable trade-off: **higher memory usage**, but significantly **faster inference** and **better accuracy**.
>
> ### **Choice of Deep Set Aggregation in the Encoder**
>
> We fully agree with the reviewer that a message-passing-based encoder may appear more expressive. In fact, we **experimented with message-passing architectures** during development. However, we found that they offered **no significant advantage** over the **simpler and more efficient Deep Set encoder**.
>
> We hypothesize that this is due to the **global nature** of the latent variable being inferred (material parameters like Young’s modulus and Poisson’s ratio are **global properties)**. Deep Sets, especially when provided with **node coordinates and features**, appear sufficient for capturing these latent structures. In contrast, message passing typically excels at modeling **localized spatial interactions**, which may be unnecessary for globally inferred variables.
>
> ### **Memory Complexity and Scalability of MaNGO**
>
> MaNGO’s memory consumption scales **linearly with the number of predicted future time steps**, due to the retention of intermediate activations:
>
> Memory(MaNGO)≈Memory(MGN)×Number of predicted steps
>
> While this can increase GPU memory usage relative to step-by-step models, it allows for **highly parallelized inference**, offering substantial gains in runtime performance.
>
> ### **Practical Memory Limitations**
>
> There is no sharp "tipping point" in MaNGO’s memory profile. The practical limit depends on the **available computing resources** and the **length of the trajectory** to be predicted. In high-resolution or long-horizon simulations, the memory cost may become non-negligible.
>
> To address this, a **hybrid decoding strategy** can be adopted: instead of predicting the entire trajectory at once, the model predicts **shorter temporal windows** (e.g., 10–20 steps) in sequence. This reduces memory consumption without reverting to fully autoregressive decoding and is a promising direction for future extensions.
>
> ### **Experimental Setup and Data Partitioning**
>
> We thank the reviewer for this important question and are happy to clarify our meta-learning protocol:
>
> - We **randomly sample material parameters** (e.g., Young’s modulus, Poisson’s ratio) from a predefined distribution to define each task.
> - For each material setting, we generate **16 simulation trials** with diverse **initial conditions** (e.g., varying object positions or velocities).
> - We apply a **random split over material parameters** to form the **meta-training and meta-testing sets**. Thus, **material properties in the test set are never seen during training**.
> - During meta-testing:
>     - We select **1 to 8 trials** from a test task to serve as the **context set**.
>     - We evaluate the model on the remaining **8 unseen trials**, ensuring a clean separation between **training**, **context**, and **evaluation**.
>
> This setup ensures **no data leakage** and measures the model's ability to **adapt to entirely new material parameters** with minimal supervision.
>
> Please let us know if additional clarification or detail would be helpful. We greatly appreciate the reviewer’s engagement and insights.

---

> ### Author Response · Authors · 2025-08-07
>
> We thank the reviewer again for the thoughtful and constructive feedback.
>
> We hope our responses have addressed the reviewer's concerns. Given the limited discussion period, we would appreciate it if the reviewer could briefly confirm whether any issues remain unresolved. We would be glad to provide further clarification if requested.
>
> We thank the reviewer for their consideration and support.

---

### Official Review · Reviewer_nqEU · 2025-07-03

**Clarity:** 4
**Significance:** 3
**Originality:** 3
**Rating:** 4
**Confidence:** 4

**Summary:**

The paper introduces Meta Neural Graph Operator (MaNGO), a meta-learned graph–network simulator that i) encodes a set of short "context" roll-outs with a spatio-temporal Conditional Neural Process (CNP) encoder, producing a latent code $r$ that captures hidden physical parameters, and ii) decodes an entire future trajectory in a single shot via a new neural-operator-style architecture that alternates spatial message passing and 1-D temporal convolutions. Experiments on four synthetic dynamics datasets show that with as few as 2-3 context simulations MaNGO achieves rollout MSE close to an oracle that knows the ground-truth material constants at test time.

**Questions:**

1) Could you report concrete numbers for MaNGO’s inference‐time GPU memory, and wall-clock latency on the Sphere Cloth Coupling (SCC) benchmark, and compare it with MGN?
2) Do you have preliminary results on larger (>10 k node) or variable-topology meshes, e.g., fracture or contact-break scenarios? What modifications, if any, would MaNGO require in those settings?
3) What is the quantitative error (e.g., RMSE) of the regressor $f_\psi(r)$ when recovering hidden material parameters such as Young’s modulus and Poisson’s ratio on held-out tasks?

**Ethical Concerns:**

["NO or VERY MINOR ethics concerns only"]

**Final Justification:**

Despite some practical limitations (notably the high training budget), this submission makes a clear and useful contribution: a well-motivated meta-learning formulation for graph simulators and a novel decoder.

I therefore maintain my original positive score, and support acceptance of this paper.

**Limitations:**

Yes.

**Paper Formatting Concerns:**

The paper follows the NeurIPS 2025 formatting instructions.

**Quality:**

4

**Strengths And Weaknesses:**

**Strengths**

- **Thorough Experiments (Quality and Significance):** The authors compare against non-meta and oracle baselines, vary the number of context runs, test robustness to noise and drop-out, and explain why ENGO, a strong equivariant baseline, fails on the Planar Bending dataset. This makes the empirical case convincing.

- **Clarity:** The paper is well-written and easy to follow.

- **Well-motivated Problem (Significance):** The paper tackles an important problem: how to make learned simulators handle new material or boundary settings without retraining (which can be computationally very costly).

- **Fresh Meta-learning Angle (Originality):** Treating the task as a Conditional Neural Process (CNP) lets the model learn from just a handful of "context” simulations, and even predict the hidden physical parameters at the same time.

- **Practical Decoder Design (Originality):** The new decoder removes the heavy SE(3)-equivariant layers equivariant layers, and replaces them with message passing networks and equivariant convolutions with a 1D CNN. It still predicts the entire trajectory in one shot and consistently outperforms the previous baselines on the reported benchmarks.


**Weaknesses**

- **Clarity:** This is minor but I believe "Figure 5" in line 290 should actually be "Figure 6".

- **All synthetic data (Quality, Significance):** Every benchmark comes from a simulator, uses a fixed graph, and has at most $\approx$ 500 nodes. There is no evidence the method works on larger meshes, changing topologies, or real sensor measurements.

- **High Training Cost (Significance):** Roughly 8,500 GPU-hours for training plus 6,000 GPU-hours for hyper-parameter search is heavy for four mid-scale tasks, and there is no comparison to optimization-based meta-learners like MAML.

- **Fixed Edge Set Assumption (Significance):** The model requires the same graph connectivity across time and tasks, limiting use in problems with fracture, contact loss, or remeshing.

---

> ### Author Rebuttal · Authors · 2025-07-30
>
> We thank the reviewer for their thoughtful and constructive feedback. We address the raised points below:
>
> **Clarity: "Figure 5" in line 290 should actually be "Figure 6"**
>
> Thank you for pointing this out. You are correct about this typo, and we will correct it in the camera-ready version.
>
> **Synthetic Data and Limited Graph Scale**
>
> We appreciate the reviewer’s concern regarding the use of synthetic datasets and relatively small, fixed-topology graphs (≤500 nodes). This was a deliberate design choice to **systematically assess MaNGO’s meta-learning capabilities** in controlled environments where:
>
> - Ground-truth simulation parameters are available,
> - Noise levels can be precisely controlled,
> - Comparisons to oracle and ablation baselines are reproducible.
>
> We agree that **scaling to real-world data and variable topology** is an important next step. While we have not yet conducted experiments on meshes with >10k nodes or on scenarios involving fracture, contact-breaking, or remeshing, we note the following:
>
> - MaNGO’s architecture  is computationally scalable and can, in principle, be applied to significantly larger graphs. The primary challenge is the **memory footprint** incurred when predicting full trajectories in a single forward pass. To address this, a promising extension is a **hybrid decoding strategy**, where instead of predicting all remaining time steps at once, the model predicts **smaller temporal windows** (e.g., 10–20 steps) sequentially. This would reduce peak memory usage by limiting the temporal span of each prediction and retain the advantages of batched inference. Alternatively, a more engineering-focused solution using multi-GPU parallelism is also a viable direction to handle large-scale setups efficiently.
> - To support **dynamic topologies**, MaNGO could be extended to **rebuild the graph connectivity at each time window**, using dynamic neighborhood graphs (e.g., k-NN or radius-based methods). While promising, this introduces additional design challenges such as temporal consistency and remeshing dynamics, which we consider **outside the scope of the current work**, but worth exploring in future research.
>
> **Training resources**
>
> We appreciate the reviewer’s concern regarding the computational resources used in our experiments.
>
> We agree that the total training and hyperparameter tuning cost—approximately **8,500 GPU-hours** for training and **6,000 GPU-hours** for search—is considerable. Upon reviewing our pipeline, we observed that **data loading and preprocessing** could still be optimized. The GPU usage logs indicate that improving these components could further **reduce idle time and accelerate training**.
>
> That said, it's important to highlight that **Graph Network Simulators (GNS)** are generally **resource-intensive to train**, especially on mid-scale physics environments with complex interactions. This characteristic applies equally to the **baseline models**, which are also non-trivial to train.
>
> Importantly, we trained all models under **identical hardware and time constraints**, ensuring a fair comparison in terms of real-world training cost. Within those constraints, MaNGO consistently achieved **strong generalization and fast inference**, which we believe justifies the computational investment.
>
> **Inference-Time GPU Memory and Runtime on SCC Benchmark**
>
> - **Runtime efficiency:** One of MaNGO’s key advantages is its ability to predict **entire trajectories in a single forward pass**, enabling effective batched inference over time. This leads to a significant **speed-up at inference time** compared to traditional **autoregressive next-step simulators**.
>
>     On the **Sphere-Cloth Coupling** benchmark:
>
>     - The **CNP encoder** takes approximately **6 ms** to compute the latent representation $r$,
>     - The **MaNGO neural operator decoder** simulates the full trajectory in just **13 ms**,
>     - In contrast, the **MGN next-step simulator** requires **approximately 500 ms**, as it predicts step-by-step over time.
> - **GPU Memory**: Memory usage scales **linearly with the number of predicted time steps**, as the model retains intermediate activations across the temporal dimension. In our experiments: Memory(MaNGO)≈Memory(MGN)×Number of predicted steps
>
>     While this can become a bottleneck for long sequences or large meshes, the **hybrid decoding strategy** discussed above offers a clear path forward, without requiring major architectural changes.
>
> Once again, we sincerely thank the reviewer for their thoughtful and encouraging feedback. We appreciate the opportunity to clarify key aspects of our work and are glad that the contributions of MaNGO were well recognized.

---

> ### Comment · Reviewer_nqEU · 2025-08-05
>
> Dear Authors,
>
> Thank you for addressing my concerns in detail.
>
> I appreciate the rationale for starting with carefully controlled, fixed-topology benchmarks and agree that the proposed hybrid-window decoder and dynamic-neighbourhood graphs are promising avenues for scaling. Your clarification that memory, rather than graph operations, is the principal bottleneck is also persuasive.
>
> That said, the substantial training time—roughly 8.5 k GPU-hours for training plus 6 k GPU-hours for hyper-parameter search—remains a significant practical limitation. Until this cost can be reduced or better justified, it tempers the immediate applicability of the method. **Nevertheless**, I find the conceptual contribution both novel and compelling, and I look forward to seeing MaNGO further developed and applied to broader settings.
>
> Consequently, I maintain my overall recommendation of "borderline accept" and leave my category scores unchanged.

---

> > ### Author Response · Authors · 2025-08-07
> >
> > We thank the reviewer for the valuable feedback and continued discussion.
> >
> > We appreciate the positive assessment of our experimental setup and clarifications.
> >
> > We want to highlight that the 8.5k GPU hours for training upper bounds all experiments presented in the paper. This includes MaNGO, its ablations and all baselines, all of which are repeated for five seeds for each of the four tasks. Each individual training takes up at most 48 hours on one GPU. For MaNGO in particular, training often converges after about 24-32 hours, which is on par with other Graph Network Simulators. We are happy to continue the discussion if there are further questions.

---

### Official Review · Reviewer_maMF · 2025-07-03

**Clarity:** 4
**Significance:** 4
**Originality:** 3
**Rating:** 5
**Confidence:** 2

**Summary:**

The paper introduces a meta-learning framework that enables Graph Network Simulators to easily adapt to unseen physical parameters without requiring retraining or explicit parameter input at test time. By leveraging Conditional Neural Processes, the proposed method encodes a small set of observed simulations into a latent representation that captures the underlying material behavior. MaNGO outperforms existing GNS baselines across several physics simulation benchmarks, achieving performance close to models with access to simulation parameters.

**Questions:**

- Could the authors provide some insights and comparisons regarding the number of parameters and runtimes?

**Ethical Concerns:**

["NO or VERY MINOR ethics concerns only"]

**Final Justification:**

The authors provided an adequate rebuttal, addressing most of the concerns raised by all the reviewers. I maintain my original score.

**Limitations:**

Yes.

**Paper Formatting Concerns:**

No.

**Quality:**

4

**Strengths And Weaknesses:**

## Strengths

- Well written paper that provides detailed theoretical analysis of the task and the method. The figures were also very helpful.

- The authors propose a novel method for critical scenarios where simulation parameters are unknown or unavailable at test time.


- The experimental section is quite rich, demonstrating consistently great results and clearly proving  architectural improvements.

- The method showcases strong generalization to unseen parameters, achieving near-oracle accuracy.


## Weaknesses

- Compared to autoregressive methods, neural operator methods might be more memory-intensive.

---

> ### Author Rebuttal · Authors · 2025-07-30
>
> We thank the reviewer for the positive feedback and their interest in our work. The reviewer asked about the **parameter count** and **runtime efficiency**, which we are happy to clarify below.
>
> - **Parameter Counts**: All models use the same Graph Neural Network (GNN) backbone to ensure a fair architectural comparison. The full network comprises approximately **3 million parameters**.
> - **Runtime Efficiency**: A key advantage of MaNGO is its ability to predict **entire trajectories in a single forward pass**, enabling **efficient batched inference** over time. This results in significant inference-time speed-ups compared to traditional **autoregressive next-step simulators**.
>
>     On the **Sphere-Cloth Coupling** benchmark:
>
>     - The **CNP encoder** takes approximately **6 ms** to compute the latent representation.
>     - The **MaNGO decoder** simulates the full trajectory in just **13 ms**.
>     - In contrast, the **MGN next-step simulator** requires approximately **500 ms**, as it predicts one step at a time.
>
> All models were trained with the **same computing resources**  to ensure a fair comparison.
>
> These results demonstrate that, although neural operator approaches like MaNGO incur a higher memory footprint (due to storing temporal activations), they offer **substantial improvements in inference speed**, making them highly suitable for **applications requiring fast rollout generation**.
>
> Once again, we thank the reviewer for their time and are happy to provide further clarifications if needed.

---

> > ### Comment · Reviewer_maMF · 2025-08-05
> >
> > I thank the authors for addressing my questions. I will maintain my original score.

---

> > > ### Author Response · Authors · 2025-08-07
> > >
> > > We thank the reviewer once again for their positive review. We are glad that the additional clarifications resolved the reviewer's questions. We appreciate the continued support for the submission.

---

### Note · Authors · 2025-08-15

We thank the reviewers and AC for their constructive engagement throughout the review process. The discussion phase allowed us to clarify key design choices and highlight additional experimental insights that strengthen the contributions of MaNGO. Below we summarize the main points addressed:

**Runtime and efficiency**: We provided detailed comparisons of parameter counts, GPU memory, and latency, showing that MaNGO predicts entire trajectories in a single forward pass (13 ms on SCC) versus 500 ms for step-by-step simulators, while maintaining a comparable model size (~3 M parameters).

**Scalability and memory trade-offs**: We highlighted that MaNGO’s architecture scales effectively to longer trajectories, with memory usage as the primary factor to manage. We proposed a hybrid-window decoding strategy that preserves MaNGO’s fast, non-autoregressive inference while enabling efficient handling of extended rollouts.

**Comparison to baselines**: We confirmed that MaNGO consistently outperforms a meta-learning MGN baseline across all benchmarks. In follow-up discussion, we also evaluated a 2-stage parameter-regression baseline, finding MaNGO’s decoder still superior while offering advantages for semi-supervised or real-world scenarios.

**Experimental setup clarity**: We described the meta-training/meta-testing protocol in detail, ensuring no data leakage and fair evaluation on unseen material parameters.

We believe these clarifications and additional experiments address the main reviewer concerns, reinforcing the novelty, practicality, and strong empirical performance of MaNGO. We look forward to incorporating these improvements in the final revision and to further extending the method to large-scale and real-world scenarios.

---

### Decision · Program_Chairs · 2025-09-17

**Decision:**

Accept (poster)

**Comment:**

This paper proposes a meta-learned graph simulator that infers (a latent representation of) hidden physical parameters from a few context rollouts. It predicts full future trajectories in a single pass using a non-autoregressive neural-operator decoder that interleaves message passing and one-dimensional temporal convolutions. Reviewer maMF praised the clarity, strong generalization to unseen parameters, and near-oracle accuracy, and asked about efficiency; the authors provided concrete comparisons and a clear account of model size and latency. Reviewer nqEU highlighted the thorough evaluation and practical decoder design, while noting limits: synthetic fixed-topology graphs, substantial training budget, and the fixed edge set; these concerns are reasonable but do not undercut the central contribution, and the authors outlined a windowed decoding path for scaling and discussed dynamic neighborhoods. Reviewer wdos emphasized the importance of the problem and requested broader meta-learning baselines (e.g., Model-Agnostic Meta-Learning, MAML) and memory scaling details; the latter were addressed convincingly, though a MAML comparison remains to be added. Reviewer EtEt asked for a simple two-stage parameter-regression baseline and more ablations; the authors report running this finding that MaNGO still work better (and argue, convincingly, that it is simpler). I think that this paper contains several interesting ideas likely to be of considerable interest to the NeurIPS community working on neural operators and simulation surrogates. Execution could be improved, in particular with more realistic problems (and problem sizes)—on the other hand, already the current setup took a significant amount of compute to get all the results for all baselines, with some error bars. For the final version, please do include the two-stage baseline and, if feasible, an optimization-based meta-learner, and briefly expand the prose discussion of efficiency, memory/trajectory trade-offs, and handling changing graph topology.